# Involvement of MicroRNA-27a-3p in the Licorice-Induced Alteration of Cd28 Expression in Mice

**DOI:** 10.3390/genes13071143

**Published:** 2022-06-25

**Authors:** Gang Feng, Guozheng Liang, Yaqian Zhang, Jicong Hu, Chuandong Zhou, Jiawen Li, Wenfeng Zhang, Han Shen, Fenglin Wu, Changli Tao, Yan Liu, Hongwei Shao

**Affiliations:** 1Guangdong Province Key Laboratory for Biotechnology Drug Candidate, School of Life Science and Biopharmaceutics, Guangdong Pharmaceutical University, Guangzhou 510006, China; fg201410405147@163.com (G.F.); lgz20220606@163.com (G.L.); 18819321381@163.com (Y.Z.); hhhujicong@163.com (J.H.); chuandong1202@163.com (C.Z.); gavin233333@163.com (J.L.); zwfsnowdream@126.com (W.Z.); shenhanbc@163.com (H.S.); trustme3344@126.com (F.W.); taochangli@126.com (C.T.); 2College of Electronic Engineering, South China Agricultural University, Guangzhou 510642, China

**Keywords:** licorice, gene expression, regulation, miRNA, *Cd28*, *mmu-miR-27a-3p*

## Abstract

Licorice has previously been shown to affect gene expression in cells; however, the underlying mechanisms remain to be clarified. We analyzed the microRNA expression profile of serum from mice treated by gavage with licorice decoction, and obtained 11 differentially expressed microRNAs (DEmiRNAs). We also screened differentially expressed genes (DEgenes) based on RNA-Seq data, and 271 common genes were identified by intersection analysis of the predicted target genes of 11 DEmiRNAs and the DEgenes. The miRNA–gene network showed that most of the hub genes were immune-related. KEGG enrichment analysis of the 271 genes identified three significant pathways, and the 21 genes involved in these three pathways, and the 11 DEmiRNAs, were constructed into a miRNA pathway–target gene network, in which *mmu-miR-27a-3p* stood out. Compared to ImmPort, there were 13 immune genes within the above group of 21 genes, and three intersected with the *mmu-miR-27a-3p* predicted target genes, *Cd28*, *Grap2* and *Cxcl12*, of which the expression of *Cd28* changed most significantly. We confirmed the regulation of *Cd28* by *mmu-miR-27a-3p* using a dual-luciferase assay, and further confirmed that overexpression of *mmu-miR-27a-3p* could significantly downregulate the expression of *Cd28* in lymphocytes. These results indicate that *mmu-miR-27a-3p* could be involved in the licorice-mediated regulation of the expression of *Cd28* in mice.

## 1. Introduction

Licorice (*Glycyrrhiza uralensis Fisch*) is one of the oldest and most frequently used medicinal plants in both the East and the West [1]. Classified as a form of tonic Chinese herbal medicine, licorice is used to treat splenic dysfunction, stomach weakness, fatigue, lack of strength, palpitation, dyspnea, cough, profuse sputum, acute pain in the abdominal cavity, limb contracture, carbuncles, and sores, as well as to alleviate drug toxicity [2,3,4,5,6]. Studies have shown that licorice has many pharmacological effects, and can exert anti-inflammatory, antiviral, and antitumor activities, as well as function in immune regulation [7,8,9,10,11,12,13,14]. 

It was reported that licorice can be used to mediate apoptosis and oxidative stress by regulating the *p-STAT3* signaling pathway in the kidney, thereby alleviating the nephrotoxicity caused by strychnine [15]. Licorice can also promote the apoptosis of some cancer cells, inhibit their growth and reproduction, and reverse the drug resistance of *ABCG2*-mediated drug-resistant cancer cells in a concentration-dependent manner [16]. Furthermore, it was shown that licorice can induce apoptosis and improve cell metabolism by regulating the expression of miRNA in cancer cells, which indicates the potential for its use in the prevention and treatment of cancer [17]. In addition, liquiritigenin, an active compound found in licorice roots, can inhibit *IL-1β*-induced inflammation and cartilage matrix degradation in rat chondrocytes [18]. Glycyrrhetinic acid induces Sirtuin 6 (*Sirt6*) overexpression by inhibiting the translocation of HMGB1 protein from the nucleus to the cytoplasm in human nasal epithelial cells, and reduces its extracellular accumulation. Thus, *Sirt6* plays an important role in maintaining nasal mucosal homeostasis, suggesting that this compound may be useful for the treatment of chronic rhinosinusitis associated with nasal polyps [19]. Together, these studies indicate that licorice may significantly affect gene expression in the body.

MicroRNAs (miRNAs) are endogenous small non-coding RNA molecules composed of 21–24 nucleotides that regulate mRNA expression by recognizing homologous sequences and interfering with the transcription, translation, and epigenetic processes of protein-coding genes [20,21,22]. As one of the common regulators of gene expression, miRNAs participate in many biological processes, such as cell proliferation, differentiation, apoptosis, growth, and development [23,24]. At present, increasing evidence shows active components of traditional herbal medicine can regulate gene expression through miRNAs, underscoring the need to study these molecules for their potential mechanism of action in herbal medicines [25,26]. 

Here we report the effect of licorice decoction on the miRNA expression profile in mice, and further confirm the effect of licorice on the immune status of mice by exploring the regulatory relationship between *mmu-miRNA-27a-3p* and *Cd28.*

## 2. Materials and Methods

### 2.1. Preparation of Licorice Decoction

Considering that water decoction is the traditional processing method of Chinese herbal medicine, in order to better simulate the therapeutic effect of licorice in practice, decoction was prepared to treat mice. Radix Glycyrrhizae Preparata (25 g) was incubated in 100 mL pure water and soaked at room temperature for 20 min, then boiled and kept at 60 °C for 2 h after boiling. Supernatant was then collected and the sediment was resuspended in 100 mL pure water, the above decocting steps repeated, and supernatant collected again. The supernatants were combined and concentrated to 25 mL with a rotary evaporator (90 °C, 0.08 MP, 14 r/min) to prepare the licorice decoction at a concentration of 1 g/mL.

### 2.2. Animal Experiments

All experiments using mice were performed using protocols approved by the Guangdong Pharmaceutical University Institutional Animal Care and Use Committee. Specific pathogen-free (SPF) BALB/c male mice (Animal Experiment Center, Guangzhou University of Traditional Chinese Medicine, Guangzhou, China), 7–8 weeks old (about 18–22 g), were maintained in an SPF environment with a 12 h light/dark cycle, and provided with autoclaved mouse chow and water ad libitum for the duration of the experiment. After 3 days of adaptive feeding, mice in the experimental group were given 0.2 mL/d licorice decoction by gavage for 7 days, and the mice in the control group were given normal saline under the same conditions. There were no less than three mice in each group. At 24 h after the last gavage, mice were sacrificed and the tissue samples were collected.

### 2.3. MiRNA Expression Profile

The total RNA of mouse serum was extracted using TRIzol and subjected to small RNA sequencing (sRNA-seq) analysis (Guangzhou Ribo, Guangzhou, China). TMM (trimmed mean of M-values) in the BioConductor Edger software package (Edger installation package for R language) was used to normalize the miRNA expression and perform statistical analysis to identify differentially expressed miRNAs (DEmiRNAs). The screening criteria were |log(fold change)| ≥ 1.5, *p*-value ≤ 0.01, and “ggplot2” was used to perform data visualization. 

### 2.4. Transcriptome Analysis

Total RNA was extracted from mouse small intestine using TRIzol and subjected to RNA-seq analysis (Novogene, Beijing, China). TMM (trimmed mean of M-values) in the BioConductor Edger software package was used to normalize gene expression and perform statistical analysis to identify the differentially expressed genes (DEgenes). The screening criteria were |log(fold change)| ≥ 1.5, *p*-value ≤ 0.01. The STRING 11.5 online tool was used to analyze the interaction network of DEgenes with a confidence level of 0.95, and the network was imported into Cytoscape3.9.0 software to further identify the core genes using the Cytohubba plugin. The DEgenes were subjected to gene ontology (GO) and Kyoto encyclopedia of genes and genomes (KEGG) enrichment analysis using the R packages “clusterProfiler” and “pathview”.

### 2.5. MiRNA Target Gene Prediction

The multiMiR R package containing eight major databases (DIANA-microT, ElMMo, MicroCosm, miRanda, miRDB, PicTar, PITA, TargetScan) was used to predict the miRNA target genes. Meanwhile, miRWalk3.0 (http://mirwalk.umm.uni-heidelberg.de/, accessed on 22 March 2021) was used for online prediction of miRNA target genes, and the results were compared with the R package to screen for common target genes.

### 2.6. Identification of Key miRNAs 

The predicted target genes were intersected with the DEgenes, and the common genes were subjected to GO and KEGG enrichment analysis as above. Genes from significantly enriched KEGG pathways and DEmiRNAs were used to construct the miRNA–gene network with Cytoscape software (v3.9.0). Genes with expression levels negatively correlated with DEmiRNA were selected and counted, and the DEmiRNAs exhibiting regulatory relationships with several of these genes were selected as candidate target miRNAs. In addition, the screening criteria of DEmiRNAs were further improved to reduce the number of DEmiRNAs, which were then compared with the above candidate target miRNAs.

### 2.7. Identification of mmu-miR-27a-3p Target Genes

The genes significantly enriched in KEGG pathways were compared with the immune-related genes from ImmPort (https://www.immport.org/home, accessed on 9 April 2021), and the common genes were then compared with the predicted target genes of *mmu-miR-27a-3p*. The miRNA binding sites on target genes were predicted using starBase 3.0 (http://starbase.sysu.edu.cn/index.php, accessed on 9 April 2021).

### 2.8. Construction of Dual-Luciferase Reporter Plasmid

The miRNA binding site within the *Cd28* 3′UTR was predicted using starBase 3.0 (http://starbase.sysu.edu.cn/index.php, accessed on 9 April 2021), and primers were designed accordingly (Appendix A). As a control, a mutant *Cd28* 3′UTR was prepared by replacing the sequence of the miRNA binding site by overlap PCR with mutation primers (Appendix A). After PCR amplification and sequencing verification, either wild-type or mutant *Cd28* 3′UTR were inserted between the *Xho I* and *Not I* sites of psiCHECK-2.

### 2.9. Cell Culture and Transfection

HEK-293T cells (purchased from ATCC) were cultured in DMEM containing 10% fetal bovine serum (FBS) at 37 °C under 5% CO_2_. Freshly isolated spleens from C57 mice were subjected to grinding, filtering, and centrifuging. Lymphocytes were then isolated from cell pellets using mouse spleen lymphocyte isolation solution (Mouse Organ Tissue Lymphocyte Separation Liquid KIT, Tianjin, China) according to the manufacturer’s protocol. After transfection, the splenic lymphocytes were cultured in 1640 medium containing 10% mouse autologous serum at 37 °C under 5% CO_2_.

1 × 10^6^ HEK-293T cells (100 μL) were transfected with 3 µg of the psiCHECK2 construct and 100 pmol mimics (Guangzhou Ribo, Guangzhou, China), using the NEPA21 Super Electroporator (NEPA GENE, Ichikawa, Japan) instrument. In addition, 1 × 10^6^ splenic lymphocytes (100 μL) were transfected with mimics (Guangzhou Ribo, Guangzhou, China) using NEPA21 Super Electroporator. 

### 2.10. Luciferase Reporter Assay

Cells were collected 24 h after transfection, and Renilla and Firefly luciferase activity was assessed using the TransDetect^®^ Double-Luciferase Reporter Assay Kit (TransGen Biotech, Beijing, China) according to manufacturer’s protocol. Each sample was repeated in triplicate. RL/FL ratio = Renilla luciferase reading/Firefly luciferase reading; normalized ratio = RL/FL ratio of mimic group divided by the RL/FL ratio of control group. 

### 2.11. Flow Cytometry

Splenic lymphocytes were collected 29 h after transfection and stained with anti-*Cd28* (BioLegend, San Diego, CA, USA). Each sample was repeated in triplicate. The expression of *Cd28* was detected by flow cytometry (Beckman, Brea, CA, USA).

### 2.12. Quantitative Real-Time PCR (qRT-PCR)

The *Cd28* expression level was determined by qRT-PCR using specific primers (Appendix A). The qPCR reaction was performed using Hieff^®^ qPCR SYBR Green Master Mix (Yeasen Bio, Shanghai, China). Each sample was repeated in triplicate. The relative expression level was quantified using the 2^−ΔΔCt^ method, and the *GAPDH* gene was used as the control.

### 2.13. Statistical Analysis

SPSS 26.0 statistical software was used to analyze the data. The graphs were made using R package ggplot2, Cytoscape 3.9.0, or GraphPad Prism 8. The values were compared by one-way analysis of variance with 0.05 as the minimum level of significance.

## 3. Results

### 3.1. Screening of Differentially Expressed microRNAs and Prediction of Their Target Genes

According to certain screening criteria, and the relatively large expression baseline (including at least one group of expression baseline ≥ 1000), 11 DEmiRNAs were identified, including five upregulated miRNAs and six downregulated miRNAs (Figure 1A,B). Two methods, multiMiR R package and miRWalk3.0 online tools, were used to predict the target genes of the 11 DEmiRNAs, resulting in the identification of 10,224 and 9501 candidate target genes, respectively, of which 5712 genes were shared (Figure 1C).

The shared candidate target genes were subjected to GO and KEGG enrichment analysis. The GO enrichment results show that, with respect to molecular functions (MF), target genes were mainly enriched in DNA-binding transcription activator activity, RNA polymerase II-specificity, and nucleoside-triphosphatase regulatory activity. For cellular components (CC), target genes were mainly enriched in the neuron-to-neuron synapse, postsynaptic density, or the synaptic membrane. For biological processes (BP), target genes were mainly enriched in synapse organization, dendrite development, regulation of cell growth, and axonogenesis (Figure 1D). In addition, KEGG enrichment results showed that these target genes were mainly related to the MAPK signaling pathway, axon guidance, and the Ras signaling pathway (Figure 1E).

### 3.2. Screening of DEgenes Based on RNA-Seq

According to certain screening criteria, 1398 DEgenes were identified, including 1033 upregulated genes and 365 downregulated genes (Figure 2A,B). In order to explore the interaction between the DEgenes, a protein–protein interaction (PPI) network was generated using the STRING online tool. Then, 212 core genes were identified using Cytoscape 3.9.0 software with the Cytohubba plugin. A majority of the top 20 genes were immune-related genes (Figure 2C). The GO analysis showed that the DEgenes were mainly enriched in antigen binding and immunoglobulin receptor binding, in terms of MF; immunoglobulin complex and circulating, in terms of CC; and B cell activation and immunoglobulin production, in terms of BP (Figure 2D). KEGG analysis showed that the DEgenes were mainly involved in immune-related pathways, such as cytokine–cytokine receptor interactions and cell adhesion (Figure 2E). These results show that the DEgenes identified are closely related to immune function. Therefore, the immune status of the body could be affected by licorice treatment.

### 3.3. Comprehensive Analysis of DEgenes and Predicted Target Genes of DEmiRNAs 

We compared the predicted target genes of DEmiRNAs with the DEgenes based on transcriptomes, and found 271 common genes (Figure 3A). Correlation analysis was carried out between these 271 genes and the 11 DEmiRNAs, resulting in the identification of 243 target genes with negative correlation. The miRNA–mRNA relationship was further visualized using Cytoscape 3.9.0, and scored using the Cytohubba plugin (Appendix A). Figure 3B shows the top 20 hub target genes. In order to explore the biological functions of the 271 potential target genes, GO and KEGG enrichment analyses were performed. GO enrichment analysis showed that DEgenes were mainly enriched in growth factor binding, sulfur compound binding, sialyltransferase activity, in terms of MF; intrinsic components of synaptic membranes, and intrinsic components of postsynaptic membranes, in terms of CC; and T cell activation, regulation of leukocyte cell–cell adhesion, and other processes closely related to immunity, in terms of BP. On the whole, the enriched genes are mainly involved in immune-related processes, of which the activation of T cell is the most significant (Figure 3C). In the KEGG enrichment analysis, there were only three significant enrichment pathways, cell adhesion molecules, viral protein interaction with cytokines and cytokine receptors, and T cell receptor signaling (Figure 3D). These results further indicate that licorice may exert effects through affecting immune activity.

### 3.4. Identification of mmu-miR-27a-3p as a Key miRNA and Prediction of Its Target Genes

The 21 DEgenes involved in the above three KEGG pathways were collected (Appendix A), and Cytoscape 3.9.0 was used to identify the miRNA–gene network between them and the 11 DEmiRNAs (Figure 4A). Of these, five DEmiRNAs known to exert negative regulatory effects on a number of genes were selected (Appendix A, Figure 4B). In addition, the screening criteria of DEmiRNAs were further restricted to |log(fold difference)| ≥ 2, *p* ≤ 0.01, and another five DEmiRNAs were identified (Appendix A, Figure 4B). Comparison of the two groups of DEmiRNAs resulted in the identification of one miRNA, mmu-miR-27a-3p (Figure 4B). 

Considering the prominent enrichment of genes with immune functions, the 21 DEgenes involved in the above three KEGG pathways were compared with 1541 immune-related genes derived from the ImmPort database, resulting in the identification of 13 common genes (Figure 4C, upper). These genes were compared with the 67 predicted target genes of *mmu-miRNA-27a-3p* found among the 271 common genes (Figure 3A, Appendix A), resulting in the identification of three target genes (Figure 4C, middle). Changes in the expression levels of these three genes are shown in the heat map (Figure 4C, lower). Of these, due to the significant expression changes, *Cd28* was selected for further investigation because of its regulatory relationship with *mmu-miRNA-27a-3p*. The qRT-PCR detection showed that after treatment with licorice decoction, the expression in healthy mice showed significant upregulation (Appendix A).

### 3.5. Regulation of Cd28 Gene Expression by mmu-miRNA-27a-3p

The binding site for *miRNA-27a-3p* in the *Cd28* 3′UTR was predicted using *starBase 3.0*, and a mutant binding site was designed as a control (Figure 5A). The sequences of both the *Cd28* 3′UTR and the mutant *Cd28* 3′UTR were amplified by overlap PCR (Figure 5B). The wild-type and mutant *Cd28* 3′UTR sequences were inserted into the 3′ end of Renilla Luciferase (Rluc) in the psiCHECK-2 vector, and the recombinant vectors were confirmed by sequencing (Appendix A).

HEK-293 T cells were co-transfected with reporter plasmids and the *mmu-miRNA-27a-3p* mimic or a control mimic. The luciferase assay showed a significant reduction in the RL/FL ratio in the *Cd28* 3′UTR + *miR-27a-3p* mimic transfected group (*p* < 0.01) (Figure 5C). The normalized ratio of the *Cd28* 3′UTR + *miR-27a-3p* mimics group was also significantly lower than that of mutant *Cd28* 3′UTR + *miR-27a-3p* mimics group (Figure 5D). There was no significant difference between the *psiCHECKTM-2* vector control and mutant *Cd28* 3′UTR control (*p* > 0.05). These results indicate that *mmu-miR-27a-3p* can directly regulate the expression of *Cd28*.

To confirm these findings, the *mmu-miR-27a-3p* mimic was transfected into splenic lymphocytes from C57 mice. Flow cytometry results showed that overexpression of *mmu-miR-27a-3p* reduced the expression level of *Cd28* (Figure 5E). Furthermore, qRT-PCR results also showed that *Cd28* was significantly downregulated by overexpression of *mmu-miR-27a-3p* (Figure 5F). These results verify that *mmu-miR-27a-3p* can regulate the expression of *Cd28*.

## 4. Discussion

Although the medicinal value of licorice has been suggested for the treatment of inflammation [27], for liver protection [28], as an antiviral [29], and as an anticancer therapeutic [30], the underlying mechanisms of these processes have not been clarified. In recent years, many studies have shown that the active components of traditional herbal medicine regulate gene expression by way of miRNAs, which may also be the mechanism through which licorice exerts its effects [31,32,33,34].

In this study, we found significant changes in the miRNA expression profile of mouse serum after treatment with licorice, including 11 miRNAs with significant differences (Figure 1B). The predicted target genes of these 11 differential miRNAs were subjected to GO enrichment analysis, and the results show the enrichment of genes with DNA-binding transcription activator activity, RNA polymerase II-specificity, involvement in neuron-to-neuron synapses, and synapse organization (Figure 1D). Thus, this implies that licorice potentially exerts comprehensive effects on the gene expression profile of mice.

Transcriptome analysis of the mouse intestine identified 1398 DEgenes following treatment with licorice, with a majority of these top 20 hub genes being immune-related (Figure 2C). GO and KEGG analysis showed that the enriched genes are known for biological functions and their involvement in pathways mainly related to the immune system (Figure 2D,E). In order to investigate the association between DEmiRNAs and DEgenes, we compared the predicted target genes of DEmiRNAs with DEgenes, and found 271 common genes. KEGG analysis of these genes identified three significantly enriched pathways (Figure 3D). The miRNA pathway–gene interaction network comparison of 21 genes involved in the three pathways, and the 11 DEmiRNAs identified five DEmiRNAs with known negative regulatory relationships with a number of genes. Furthermore, five DEmiRNAs using higher screening criteria were identified based on sRNA-seq data. As the only common miRNA between the two groups of DEmiRNAs, *mmu-miR-27a-3p* was particularly noteworthy (Figure 4B), as it reportedly regulates immune cell function [35]. Moreover, overexpression of *miR-27a-3p* can inhibit the inflammatory response during spinal cord injury, by reducing TLR4 and inhibiting the expression of *TNF-α* and *IL-6* [36]. In breast cancer, exosomal *miR-27a-3p* was used as an intervention target to inhibit *PD-L1* expression in macrophages, and to block the activation of the PTEN-AKT/PI3K pathway to inhibit immune escape in breast cancer cells [37]. In addition, it was reported that adipocyte-derived *exo-miR-27a-3p* can inhibit *ICOS+* T cell proliferation and *IFN-gamma* secretion [38]. Therefore, it is reasonable to speculate that *mmu-miR-27a-3p* could be involved in the regulation of gene expression, and especially of immune-related genes, by licorice.

To identify *mmu-miR-27a-3p* target genes, we compared the 21 genes involved in the above three pathways with 1541 immune genes derived from the ImmPort database, and found 13 common genes. These 13 genes were then compared with predicted target genes of *mmu-miRNA-27a-3p*, and three common genes, *Cd28*, *Grap2*, and *Cxcl12*, were identified (Figure 4C). Combined with the expression changes, *Cd28* was selected for further exploration of its regulation by *mmu-miR-27a-3p*.

A luciferase assay showed that *mmu-miR-27a-3p* could affect the expression of a reporter gene downstream of the *Cd28* 3′UTR (Figure 5C,D). Furthermore, both flow cytometry and qRT-PCR showed that overexpression of *mmu-miR-27a-3p* in the splenic lymphocytes of C57 mice could significantly downregulate the expression of *Cd28* (Figure 5E,F). These results confirm that *Cd28* is a target gene of *mmu-miR-27a-3p*.

It is well known that *Cd28* plays a pivotal role in T cell activation. In addition, *Cd28* is necessary for humoral responses to viral infection, vaccination, and in allergy models [39,40,41]. It was reported that *Cd28* signaling in plasma cells (PC) is essential for promoting plasma cell survival and sustaining antibody titers [42,43]. *Cd28* also promotes the functional maturation of *NKT* cells and the development of innate-like *CD8*+ *T* cells [44]. Boesteanu et al. showed that both *CD4+* and *CD8+* memory T cells require *Cd28* co-stimulation for maximal expansion and pathogen clearance [45]. However, it was reported that the *B7/Cd28/CTLA4* pathway has the ability to both positively and negatively regulate immune responses [46]. Moreover, *Cd28* plays a crucial role in the maintenance of regulatory T cell (Treg) pool size through promoting the development and proliferation of these cells [47]. *B7-Cd28* co-stimulation modulates central tolerance via thymic clonal deletion and Treg generation [48]. Interestingly, it was reported that glycyrrhizic acid can regulate T and B cell proliferation, and improve cognition in aging mice [49]. Furthermore, it was reported that glycyrrhizin may reverse the increased susceptibility of thermally injured mice to HSV infection through the induction of *CD4+ CD28+* contrasuppressor T cells [50], and that glycyrrhizic acid can upregulate *Foxp3* and enhance the activity of Treg cells [51,52]. Therefore, further study of the role of *mmu-miR-27a-3p* and *Cd28* in the regulation of immune responses following treatment with licorice is warranted. 

## 5. Conclusions

In conclusion, our findings reveal that treatment with licorice can significantly alter the gene expression profile of mice and affect the expression of both miRNAs and genes. Moreover, we identify a regulatory relationship between *mmu-miR-27a-3p* and *Cd28*, which lays the foundation for further elucidating the mechanism of action of licorice. 

## Figures and Tables

**Figure 1 genes-13-01143-f001:**
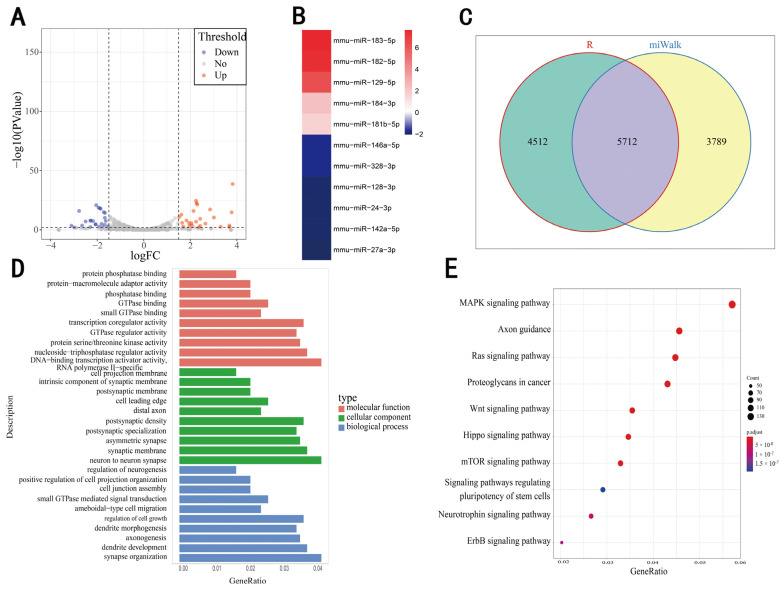
Screening of DEmiRNAs and prediction of target genes. (**A**) Volcano plot of differential miRNAs with |log(fold change)| ≥ 1.5, *p* ≤ 0.01 as screening criteria. (**B**) Expression changes of 11 DEmiRNAs. (**C**) The predicted target genes of 11 DEmiRNAs using multiMiR R package and miRWalk 3.0 online tools. (**D**) GO enrichment of common target genes using two prediction methods. (**E**) KEGG enrichment of common target genes using two prediction methods.

**Figure 2 genes-13-01143-f002:**
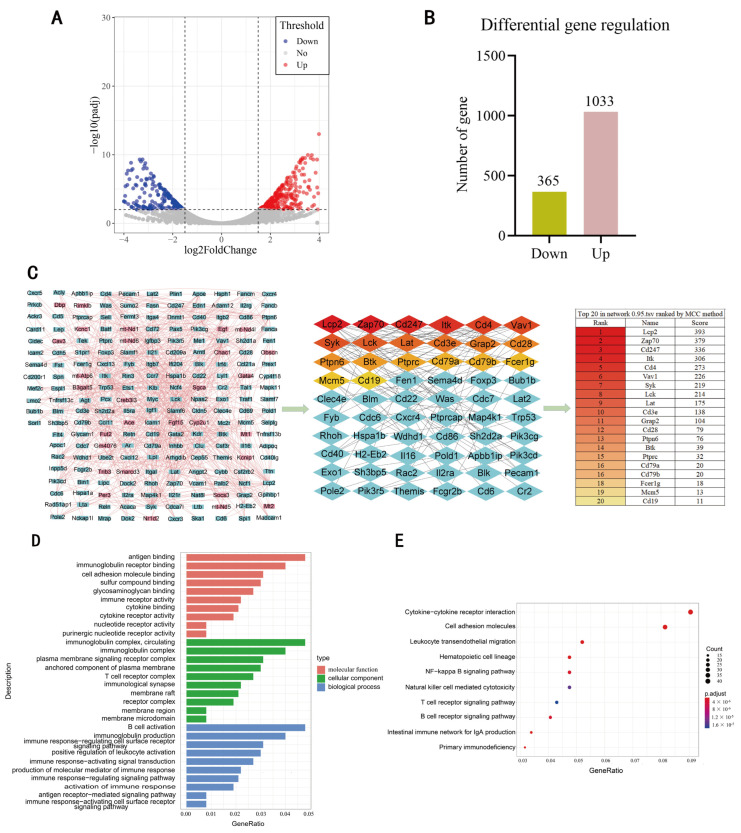
Screening of DEgenes based on RNA-Seq. (**A**) Volcano plot of differential miRNAs with |log (fold change)| ≥ 1.5, *p* ≤ 0.01 as screening criteria. (**B**) Distribution of upregulated and downregulated DEgenes. (**C**) The interaction network between DEgenes generated using the STRING online tool and visualized with Cytoscape 3.9.0, in which the core genes were identified using Cytohubba plugin. (**D**) GO enrichment of DEgenes. (**E**) KEGG enrichment of DEgenes.

**Figure 3 genes-13-01143-f003:**
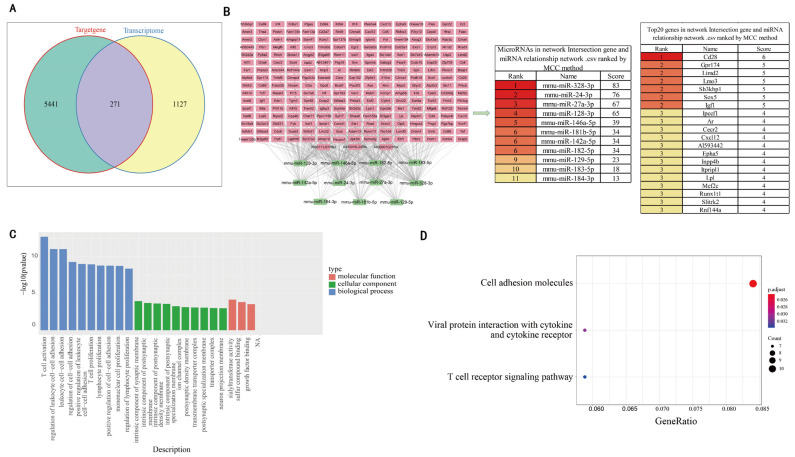
Comparative analysis of DEgenes and the predicted target genes of 11 DEmiRNAs. (**A**) The intersection between DEgenes and the predicted target genes of 11 DEmiRNAs. (**B**) Correlation analysis between 271 common genes and 11 DEmiRNAs identified 243 negatively correlated genes. The miRNA–mRNA relationship was visualized with Cytoscape 3.9.0, in which the core genes were identified using the Cytohubba plugin. (**C**) GO enrichment of 271 common genes. (**D**) KEGG enrichment of 271 common genes.

**Figure 4 genes-13-01143-f004:**
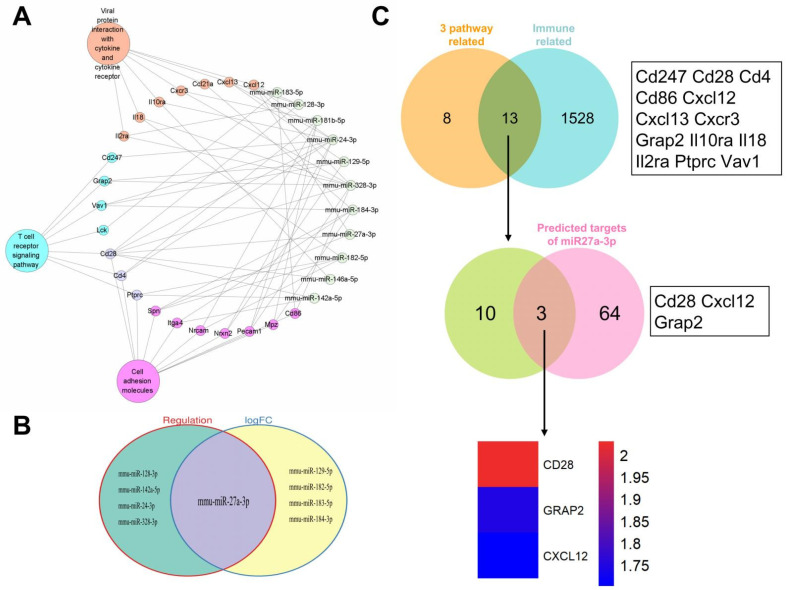
Identification of key miRNAs and target gene prediction. (**A**) The miRNA–gene interaction network between 11 DEmiRNAs and 21 DEgenes involved in the above three KEGG pathways. (**B**) The intersection between DEmiRNAs that have negative regulatory relationships with more genes in A and the DEmiRNAs identified using the stricter screening criteria (|log(fold difference)| ≥ 2, *p* ≤ 0.01). (**C**) The intersection between 21 DEgenes involved in the above three KEGG pathways and 15 immune-related genes derived from ImmPort database identified 13 common genes, which were compared with the 67 predicted target genes of *mmu-miRNA-27a-3p*; among the 271 common genes mentioned above, this resulted in the identification of three common genes. Changes in expression of the three common genes are shown.

**Figure 5 genes-13-01143-f005:**
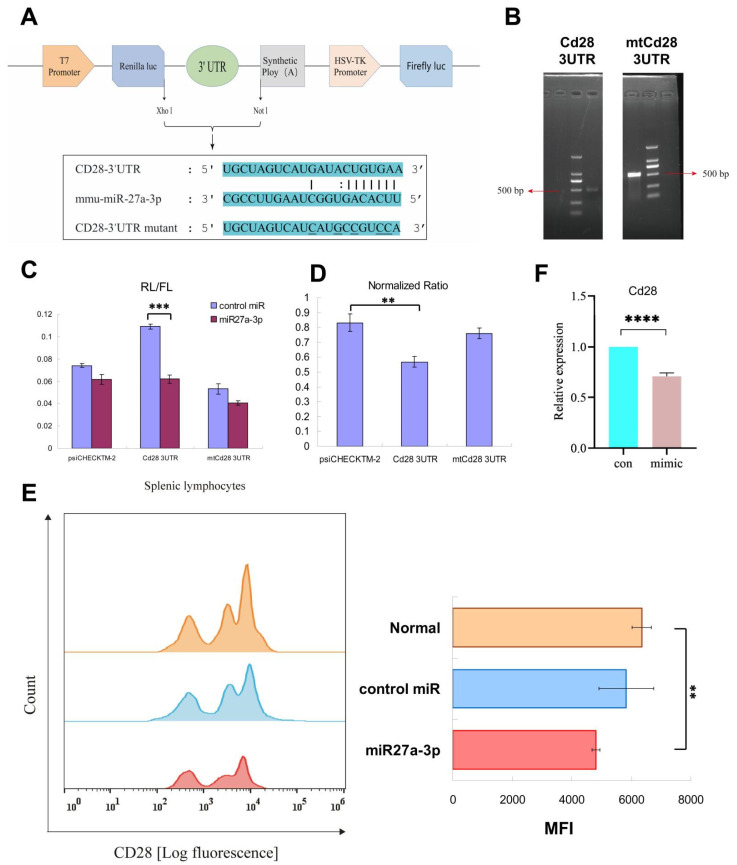
Regulation of *Cd28* gene expression by *mmu-miRNA-27a-3p*. (**A**) A *miRNA-27a-3p* binding site with the *Cd28* 3′UTR was predicted using starBase 3.0 and a mutant binding site was designed as a control. (**B**) Amplification of the *Cd28* 3′UTR and the preparation of mutant *Cd28* 3′UTR by overlap PCR. (**C**) The RL/FL ratio of each group. (**D**) The normalized ratio of each group. (**E**) Flow cytometric analysis of the expression of *Cd28* on splenic lymphocytes from C57 mice transfected with *mmu-miR-27a-3p* mimics. The ratio and mean fluorescence intensity (*MFI*) were analyzed. (**F**) qRT-PCR detection of *Cd28* in C57 mice spleen lymphocytes. **, *p* < 0.01; ***, *p* < 0.001; ****, *p* < 0.0001.

## Data Availability

Not applicable.

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
