# Peer review of "Involvement of MicroRNA-27a-3p in the Licorice-Induced Alteration of Cd28 Expression in Mice"

_genes, 2022, doi:10.3390/genes13071143_

Round 1

Reviewer 1 Report

The authors performed microRNA expression profile on the serum from licorice-fed mice. They identified differentially expressed genes and identified significantly regulated genes. They focused on one gene (mmu-miR-27a-3p) and predicted its target genes. They performed cellular experiments and confirmed mmu-miR-27a-3p downregulates Cd28 expression.

Comments:

  1. For all figures, please change legend size to make them visible. None of them are visible in this version.

  2. The biological replication numbers need to be noted. For example, for gene differential experiments, how many biological replicates were used.

  3. Does Cd28 gene expressed differently in cells or mice upon licorice active ingredient treatment or licorice decoction on mice?

Author Response

Dear Editor,

Thank you very much for your letter and the comments from the reviewers about our manuscript entitled “Involvement of MicroRNA-27a-3p in the Licorice-induced alteration of Cd28 expression in mice” (Manuscript ID: genes-1769052). We really appreciate the careful reading of our manuscript and valuable suggestions of the reviewer. We have carefully considered the comments and have revised the manuscript accordingly. The detailed responses to the comments are attached.

If you have any question about this paper, please don’t hesitate to let me know.

Best regards.

Gang Feng

Response to Reviewer 1 Comments

Reviewer #1:

1. For all figures, please change legend size to make them visible. None of them are visible in this version.

Response: Thank you for your advice. We have modified the images and changed the legend size to make them visible.

2. The biological replication numbers need to be noted. For example, for gene differential experiments, how many biological replicates were used.

Response: As suggested by the reviewer, we have supplemented the relevant biological replication information. See 2.2, 2.10, 2.11, 2.12 in the Materials and Methods section of the text.

3. Does Cd28 gene expressed differently in cells or mice upon licorice active ingredient treatment or licorice decoction on mice?

ResponseAs the reviewer suggests, we supplemented the detection of expression level of Cd28 gene in spleen and intestine of mice treated with licorice decoction, and the results showed that the licorice decoction treatment significantly increased the expression level of Cd28 gene (see Results 3.4 and Supplementary Fig. 1).

Reviewer 2 Report

Interesting research

I have a few comments: 

1. In the introduction section, the authors should clarify that the main focus of their work is investigating the effect of licorice on the immune status by examing the expression of selected miRNAs

2. In the material/method section, the authors should explain why they preferred to use licorice devotion instead of extraction

3.. In the results section, the authors should prevent the repetition of some parts of the materials/methods once again in the results section

Author Response

Dear Editor,

Thank you very much for your letter and the comments from the reviewers about our manuscript entitled “Involvement of MicroRNA-27a-3p in the Licorice-induced alteration of Cd28 expression in mice” (Manuscript ID: genes-1769052). We really appreciate the careful reading of our manuscript and valuable suggestions of the reviewer. We have carefully considered the comments and have revised the manuscript accordingly. The detailed responses to the comments are attached.

If you have any question about this paper, please don’t hesitate to let me know.

Best regards.

Gang Feng

Response to Reviewer 2 Comments

Reviewer #2:

1. In the introduction section, the authors should clarify that the main focus of their work is investigating the effect of licorice on the immune status by examing the expression of selected miRNAs

Response: We agree with the reviewer’s suggestion and the changes have been made, the last paragraph of the introduction has been changed to read: Here we report the effect of licorice decoction on the miRNA expression profile in mouse, and further confirm the effect of licorice on the immune status of mice by exploring the regulatory relationship between mmu-miRNA-27a-3p and CD28.

2. In the material/method section, the authors should explain why they preferred to use licorice devotion instead of extraction

Response: Water decoction is the traditional processing method of Chinese herbal medicine, and we simulated this traditional processing method to treat mice. The corresponding explanations have also been made in the article (see Materials and Methods 2.1).

3. In the results section, the authors should prevent the repetition of some parts of the materials/methods once again in the results section

Response: As suggested by the reviewers, the changes have been made (see Results 3.1, 3.2).
